# Protective Effect of Alpha-Lipoic Acid on Salivary Dysfunction in a Mouse Model of Radioiodine Therapy-Induced Sialoadenitis

**DOI:** 10.3390/ijms21114136

**Published:** 2020-06-10

**Authors:** Jung Hwa Jung, Jin Hyun Kim, Myeong Hee Jung, Seung Won Kim, Bae Kwon Jeong, Seung Hoon Woo

**Affiliations:** 1Department of Internal medicine, Gyeongsang National University School of Medicine and Gyeongsang National University Hospital, Jinju 52727, Korea; jhring@daum.net; 2Institute of Health Science, Gyeongsang National University School of Medicine, Jinju 52727, Korea; ajini7044@hanmail.net (J.H.K.); blue129j@hanmail.net (B.K.J.); 3Biomedical Research Institute, Gyeongsang National University Hospital, Jinju 52727, Korea; yallang7@daum.net; 4Animal Molecular Imaging Team, Research Core Center, National Cancer Center, Goyang 10408, Korea; seungwonkim27@ncc.re.kr; 5Department of Radiation Oncology, Gyeongsang National University School of Medicine and Gyeongsang National University Hospital, Jinju 52727, Korea; 6Department of Otorhinolaryngology-Head and Neck Surgery, Dankook University, Cheonan 31116, Korea

**Keywords:** alpha lipoic acid, Thyroid, RI, complications, salivary gland

## Abstract

Radioiodine (RI) therapy is known to cause salivary gland (SG) dysfunction. The effects of antioxidants on RI-induced SG damage have not been well described. This study was performed to investigate the radioprotective effects of alpha lipoic acid (ALA) administered prior to RI therapy in a mouse model of RI-induced sialadenitis. Four-week-old female C57BL/6 mice were divided into four groups (*n* = 10 per group): group I, normal control; group II, ALA alone (100 mg/kg); group III, RI alone (0.01 mCi/g body weight, orally); and group IV, ALA + RI (ALA at 100 mg/kg, 24 h and 30 min before RI exposure at 0.01 mCi/g body weight). The animals in these groups were divided into two subgroups and euthanized at 30 or 90 days post-RI treatment. Changes in salivary ^99m^Tc pertechnetate uptake and excretion were tracked by single-photon emission computed tomography. Salivary histological examinations and TUNEL assays were performed. The ^99m^Tc pertechnetate excretion level recovered in the ALA treatment group. Salivary epithelial (aquaporin 5) cells of the ALA + RI group were protected from RI damage. The ALA + RI group exhibited more mucin-containing parenchyma and less fibrotic tissues than the RI only group. Fewer apoptotic cells were observed in the ALA + RI group compared to the RI only group. Pretreatment with ALA before RI therapy is potentially beneficial in protecting against RI-induced salivary dysfunction.

## 1. Introduction

The standard treatment for differentiated thyroid cancer is thyroidectomy followed by high-dose radioiodine (RI) treatment to completely ablate thyroid remnants. However, due to the presence of sodium iodide symporter, iodide is actively transported into a number of non-thyroidal tissues, such as the salivary glands (SGs), stomach, lacrimal glands, and lactating mammary glands, as well as the thyroid gland [1]. Among these, SGs are known to exhibit irreversible radiation-induced damage—such as sialoadenitis, xerostomia, and neoplasia—as seen in patients with differentiated thyroid cancer after high-dose RI therapy [2,3]. Although high-dose RI treatment is regularly performed under SG stimulation to minimize SG dysfunction, this treatment has shown only limited success [4,5]. It is essential to develop methods to minimize the morbidity associated with this standard treatment protocol.

The acinar and ductal cells in the SG are damaged by RI. RI is concentrated in the ductal epithelial cells and leads to SG damage [6]. RI induces the generation of reactive oxygen species (ROS), which damage the DNA and disrupt the structure of SG cells. Free radicals in RI-treated subjects cause salivary dysfunction. Amifostine is the only drug reported to date that is capable of reducing the side effects of ionizing radiation in SGs, and the US FDA has approved its clinical use as a radioprotector [7]. However, its applicability is limited due to the associated adverse effects, including hypocalcemia, diarrhea, nausea, vomiting, sneezing, somnolence, and hypotension. In addition, the high cost of the drug makes its use difficult in the majority of patients. Although research is underway regarding the protection of SGs after RI therapy, ideal radioprotective agents have yet to be found.

Recently, it was reported that alpha lipoic acid (ALA) (Appendix A) protects against radiation-induced normal tissue injury and dysfunction [8,9,10,11]. ALA is a strong antioxidant with high reactivity to free radicals, and it elevates tissue levels of glutathione [12]. ALA has been shown to be effective in preventing pathological processes, such as ischemia–reperfusion injury [13], diabetes [14], hypertension, and radiation injury [15]. ALA is under examination for its potential as an ideal radioprotectant, and could be nontoxic, safe, readily available, and cost-effective. In addition, ALA is already in use for treatment of diabetic neuropathy.

This study was performed to evaluate the radioprotective effects of ALA on RI-induced damage to the SGs.

## 2. Results

### 2.1. ALA Ameliorates the Body Weight Loss and Impaired Saliva Secretion Induced by Radioiodine Exposure 

The mean body weight of the RI-treated mice group was significantly lower than the control group at 90 days, while the weight in the RI +ALA and ALA groups were similar to the control group (Figure 1A). The salivary flow rates in the RI + ALA treated group was higher than in the RI only group at 30 and 90 days post RI (Figure 1B). The salivary lag time improved in the RI + ALA group compared to the RI only group at 90 days post treatment (Figure 1C). 

### 2.2. Enzyme-Linked Immunosorbent Assay (ELISA) for Thyroid Function

To verify the effects of ALA on thyroid function following RI therapy, ELISAs for thyroid stimulating hormone (TSH) were performed (both 30 and 90 days). The TSH levels were significantly increased in RI only group and tend to decrease by ALA treatment but not significant (Fig 1D). Thyroid function was not significantly changed by ALA treatment in RI +ALA group compared with RI only group, so the pretreatment of ALA does not reduce RI thyroidal treatment effect.

### 2.3. ALA Decreased Structural Changes Induced by RI Therapy

Histological changes were identified by hematoxylin and eosin (HE) and Masson trichrome (MT) staining at 30 and 90 days post-RI therapy. The control group showed well-demarcated lobules with dense acini and fully developed ducts. At both 30 and 90 days post-RI, the RI only group exhibited multifocal areas of cytoplasmic vacuolization. The ALA + RI group had intact structures similar to the control group (Figure 2A–C). Diffuse fibrotic tissues were observed in the RI only group at both 30 and 90 days post-RI (Figure 2D,E). The signal for fibrosis was stronger at 90 days than at 30 days. The ALA + RI group showed less perivascular and periductal fibrosis than the RI only group (Figure 2D–F). 

### 2.4. ALA Reduces RI-Induced Salivary Apoptotic Cell Death

To determine the effect of ALA on RI-induced salivary apoptotic cell death, terminal deoxynucleotidyl transferase biotin-dUTP nick end labeling (TUNEL) assay was performed. Faint signals were found in the control and ALA only groups both 30 and 90 days post-RI therapy; these were more abundant in the acinar than in the ductal cells, and more distinct at 30 days than 90 days (Figure 3A,B). However, the positive signals were significantly increased in the RI only group compared to the control and ALA only groups. Moreover, the total number of apoptotic cells was significantly lower in the ALA + RI group compared to the RI only group, at both 30 and 90 days post-RI therapy (Figure 3C). 

### 2.5. ALA Prevents RI-Induced Cellular Senescence in the SGs

To investigate cellular senescence in RI-induced SGs, we stained SGs for the classic biomarker of senescence, senescence-associated β-galactosidase (SA-β-gal). In the RI only group, the acinar and ductal cells were significantly positive for SA-β-gal and the staining was distinctly greater at 90 days than at 30 days (Figure 3D,E). The ALA + RI group showed a significant decrease in positive signals for cellular senescence (Figure 3F).

### 2.6. ALA Ameliorates RI-Induced Reduction in AQP-5 Expression

Staining for the salivary epithelial marker, aquaporin 5 (AQP-5), revealed acini-rich SGs in the control group. In contrast, the RI only group showed a few AQP-5-positive areas at both 30 and 90 days post-RI (Figure 4A,B). The ALA + RI group showed similar staining to the normal control group (Figure 4C).

### 2.7. ALA Ameliorates RI-Induced Salivary Dysfunction

We used single-photon emission computed tomography (SPECT) to examine salivary function after RI therapy. Based on the SPECT images at 30 days post-RI, the amount of ^99m^Tc pertechnetate elimination was lower in the RI only group compared to the other groups. The degree of ^99m^Tc pertechnetate elimination significantly lesser in the RI only group at 110 and 120 min. The excretion of ^99m^Tc pertechnetate in the ALA only group was similar to the control group (Figure 5A,B). At 90 days post-RI, the level of ^99m^Tc pertechnetate elimination was significantly lower in the RI only group compared to the other groups at 70, 90, 100, 110, and 120 min. The excretion of ^99m^Tc pertechnetate in the ALA only group was better than that in the control group (Figure 5C,D). These observations indicated that ALA ameliorated salivary dysfunction after RI therapy.

## 3. Discussion

RI mainly uses ionizing radiation to produce cell death through the formation of free radicals. In thyroid cancer patients, oral RI therapy after thyroidectomy is also known to produce free radicals [16,17]. Reactive oxygen species or free radical formation post-oral RI therapy leads to a cytotoxic process that ultimately results in the death of SG cells [18]. However, in addition to intracellular antioxidants, such as glutathione, and enzymes such as glutathione transferase, reductase, peroxidase, and superoxide dismutase, certain chemical substances known to act as antioxidants can protect biological systems against radiation-induced toxicity [19,20,21]. Although many compounds have shown promising results in preventing radiation-induced injury in vitro, most failed to obtain regulatory approval at the preclinical stage due to safety concerns. 

Thiol-containing compounds have long been known to have protective effects against radiation damage [22]. Amifostine was shown to protect animals from lethal doses of irradiation based on selective radioprotection of normal tissues from the toxic effects of ionizing radiation, with no protection of malignant transformed cells, in preclinical studies [23,24]. Initial studies of patients with differentiated thyroid cancer demonstrated the radioprotective effect of amifostine on SG dysfunction following high-dose RI treatment [25]. However, limited data are available regarding the effects of amifostine on SGs, as well as on thyroid tissue. In addition, amifostine has severe adverse effects that result in discontinuation of its use in some patients [26].

The protective effects of ALA against radiation-induced tissue damage have been reported in experimental animal models [8,9,10]. ALA ameliorated radiation-induced tissue damage by decreasing oxidative stress, apoptotic cell death, inflammation, and fibrosis in the oral mucosa, small intestine, thyroid, and SG [8,9,10]. In particular, ALA is known to be involved in the recovery of reduced salivary flow and volume caused by irradiation. Few available data have been reported, indicating that antioxidants—including ALA—can protect the SGs against internal exposure to β-emitting RI. Structural changes in the SGs after RI therapy include diffuse lipomatosis, acinar cell metamorphosis, and salivary glandular ductal dilation or stenosis [7,27]. We also identified structural changes with cytoplasmic vacuolization, increased diffuse fibrosis, and lymphocyte infiltration in the RI-treated SGs. However, disintegration of ductal cells and the destruction of acinar cells were decreased in the RI + ALA group. These histological observations suggested that ALA may be useful in preventing RI-induced SG damage. To our knowledge, this is the first report demonstrating a protective effect of ALA against salivary dysfunction after RI therapy.

Apoptosis is one of the mechanisms involved in SG injury after exposure to radiation. We demonstrated previously that apoptotic cell death occurred in both acinar and ductal cells of the SGs and oral mucosa after radiation exposure, and that these effects were decreased by ALA [10]. In accordance with these data, the number of TUNEL-positive signals was significantly lower in the RI + ALA group than the RI only group. Together with apoptosis, cellular senescence is known to occur in irradiated SGs [28]. The senescent phenotype with β-gal-positive signals was significantly increased in our RI only group, but reduced in the RI + ALA group. Therefore, it appears that the radioprotective effects of ALA on RI-induced SG damage could be related to the apoptotic and senescence pathways.

For evaluation of human SG function, SG scintigraphy with ^99m^Tc pertechnetate is a standard noninvasive procedure in routine clinical practice, it is well tolerated by patients, easy to perform, and can be repeated several times [29]. The correlation between ^99m^Tc uptake into the SGs and saliva secretion has also been established [30,31]. In the present study, the ALA treatment group showed greater excretion of ^99m^Tc pertechnetate than the RI only group, suggesting a relationship with RI-induced ductal distress.

Although thyroid cancer surgery has advanced [32,33,34,35,36,37], patients with differentiated thyroid cancer undergo high-dose RI treatment to achieve complete ablation of thyroid remnants following total thyroidectomy. Therapeutic RI exposure, such as exposure to ^131^I, is known to generate oxidative stress and cause cellular damage in target organs [38]. With RI, ^131^I also accumulates at high levels in the salivary tissues through the sodium iodide symporter. Accumulation in the SGs causes irreversible damage and ultimately impairs the quality of life of patients [30,38,39,40]. We already reported the protective effects of ALA on radiation-induced the thyroid tissue in the previous study [9] and we also show that ALA can protect salivary glands in the present study. A previous publication was focused on a normal thyroid organ (big tissue). However, the RI therapy was focused on a remnant tissue, not whole thyroid or big tissue. Therefore, we think that ALA will not affect the RI therapy to the remnant thyroid tissue. 

We performed experiments in an RI ablation model without thyroidectomy in the present study. A murine thyroidectomy model can be obtained through surgical excision or RI administration. RI-mediated ablation is beneficial in that it avoids the need for surgical intervention and the potential associated complications, as well as the need for postoperative care. Moreover, murine thyroidectomy may be incomplete because of the difficulties associated with the microsurgical technique. In addition, as the SGs are located very close to the thyroid glands, surgery can have a negative effect on the function and morphology of the SGs, due to surgical damage and postoperative fibrosis. As shown in SPECT images from RI and ALA+RI group, we verified that the thyroid glands were completely ablated after RI administration. 

ALA is a potentially useful anticancer agent with a mechanism involving the inhibition of cancer cell proliferation or sensitization of cancer cells to apoptosis in various solid tumors [41,42,43]. ALA has also been shown to be useful in the chemoprevention of cancers [44,45]. However, the protective effect of ALA on normal tissue during cancer treatment is less well understood. ALA is a very safe anti-oxidant and used to treat various condition for long-time in human. There are the data suggesting long-term safety of ALA in SD and Wistar rats with high dose of 180mg and 121 mg/kg, respectively [46,47]. Although usual dose of ALA is safe, acute high-dose ingestions might be fatal. There is the case report of ALA intoxication after a total 18g of ALA with a suicidal intention [48]. Therefore, treatment dose of ALA should be administered by physician in clinical field.

In conclusion, this study demonstrated that ALA effectively protects SGs against the negative effects of RI therapy, via its beneficial effects on SG histological and functional recovery. Although further studies are required to characterize the pathways involved in the effects of ALA, it may be a useful therapeutic option to protect the SGs against damage in patients with thyroid cancer following RI therapy.

## 4. Materials and Methods

### 4.1. Animal Studies

Female C57BL/6 mice (18–22 g; Orient Bio Inc., Seongnam, Korea) were maintained under controlled temperature/light conditions in an animal house with free access to water and standard mouse diet. Animal studies were performed in compliance with the guidelines issued by the National Cancer Center Institutional Animal Ethics Committee (NCC-16-325B). Animals were divided into the following four groups (*n* = 10 animals per group): group I, normal control; group II, ALA alone (100 mg/kg; Bukwang Pharmaceutical Co., Seoul, Korea); group III, RI alone (0.01 mCi/g body weight orally, ^131^I; New Korea Industrial, Seoul, Korea); and group IV, ALA treatment at 24 h and 30 min before RI exposure (ALA + RI). We chose the dose of ALA, and frequency thereof, based on previous studies [14,15,49].

### 4.2. Measurement of Salivary Function

Salivary function was evaluated by measurement of saliva secretion. Pilocarpine (0.01 mL/g body weight i.p.; Isopto Carpine; Alcon Korea Ltd., Seoul, Korea) was injected, and after 8 min the saliva output was collected from the mouth for 5 min. After measuring total body weight of the mice, the total amount of saliva that accumulated during a period of 10 min after pilocarpine injection in fresh tubes, and the volume was normalized to body weight. Salivary lag times and flow rate were also measured. Salivary flow rates (total saliva weight divided by the collection time) and lag time (time from stimulation to the commencement of saliva secretion) were calculated. 

### 4.3. Enzyme-Linked Immunosorbent Assay (ELISA)

To verify the effects of ALA on thyroid function following RI therapy, ELISAs for TSH were performed (both 30 and 90 days). Serum were collected from all groups in all time points and stored at −80 °C. The levels of TSH (cat no. MBS2502905. MyBioSource, San Diego, CA, USA) were measured according to the manufacturer’s instructions.

### 4.4. Histopathology

Tissues were fixed in 4% paraformaldehyde in 0.1 M PBS, embedded in paraffin, and cut into 5-μm. The sections were stained with HE. Histopathological injury in HE staining has been scored by grading the number of acinar cells with cytoplasmic vacuoles under ×200 magnification field; 0, 0–1; 1, 2–5; 2, 5–10; 3, 10–15; 4, 15–20; 5, >20. For analyzing the degree of collagen deposition, sections were stained with MT (Masson’s Trichrome kit, Sigma Diagnostics, St. Louis, MO, USA). SG fields that were randomly selected at ×400 magnification were assessed in each mouse, and the density of trichrome-positive signals was analyzed using NIS-Elements BR 3.2 (Nikon, Japan).

### 4.5. Morphological Analysis of Tissues and TUNEL Assay 

SGs were immediately placed in 4% paraformaldehyde at room temperature, embedded in paraffin, and cut into 5-μm thick sections. SGs were stained with Masson’s trichrome (MT) and with hematoxylin and eosin (H&E). Apoptosis in submaxillary gland tissues was determined by TUNEL assay using an ApopTag Plus In Situ Apoptosis Kit (Chemicon Int., Temecula, CA, USA). TUNEL-positive cells were detected under ×400 magnification, and TUNEL-positive cells were counted in 10 random high-power fields. TUNEL assays were performed at 30 and 90 days post-RI exposure. The sections were visualized by light microscopy and captured and analyzed digital images. Data was analyzed by signal intensity using NIS-Elements BR 3.2 (Nikon, Japan) in 10 random fields and described as fold change. The fold changes are calculated as the ratio of the final value in each group to the value in control group at day 30 (set as “1”).

### 4.6. SA-β-Gal Staining

For detection of SA-β-gal in tissues, SGs were fixed for 15 min in 1× fixative solution at room temperature, washed twice in 1× phosphate buffered saline (PBS) and stained overnight at 37 °C using an SA-β-gal staining kit (BioVision, Mountain View, CA, USA) according to the manufacturer’s instructions. Stained tissues were observed under a microscope for development of blue color. Next, we visualized the sections by light microscopy and captured and analyzed digital images. Data was analyzed by signal intensity using NIS-Elements BR 3.2 (Nikon, Japan) in 10 random fields and described as fold change. The fold changes are calculated as the ratio of the final value in each group to the value in control group at day 30 (set as “1”).

### 4.7. SPECT Protocol for the Animal Study

At 30 and 90 days post-RI, technetium pertechnetate (55.5 MBq, [^99m^Tc]TcO_4_^–^; New Korea Industrial) was administered intraperitoneally (i.p.) to anesthetized mice maintained in an unconscious state during the entire imaging protocol using isoflurane (2 vol % in air). Whole-body single-photon emission computed tomography (SPECT) imaging was started immediately after [^99m^Tc] TcO_4_^–^ injection and repeated every 10 min for 120 min (NanoSPECT; Bioscan Inc., Washington, DC, USA). Overall, 13 images were obtained per mouse. A fresh solution of pilocarpine (0.5 mg/mL) was then prepared in PBS, and administered at 0.01 mL/g body weight (i.p.) 60 min after SPECT.

### 4.8. Whole-Body SPECT Protocol

Whole-body SPECT images were obtained using a large field-of-view rotating gamma camera equipped with four multi-pinhole collimators. The acquisition parameters were as follows: 24 projections over 360°, circular orbit, and a total acquisition time of 6 min (4 s per projection). Tomographic images were reconstructed using an iterative reconstruction algorithm [13,14].

### 4.9. Immunohistochemistry 

After deparaffinization, the sections were incubated with primary antibodies against polyclonal anti-AQP5 (Abcam, Cambridge, UK) followed by biotin-conjugated secondary IgG (diluted 1:200; Vector Laboratories, Burlingame, CA, USA), avidin–biotin–peroxidase complex (ABC Elite Kit; Vector Laboratories), and diaminobenzidine tetrahydrochloride. Next, we visualized the sections by light microscopy and captured and analyzed digital images. Data was analyzed by signal intensity using NIS-Elements BR 3.2 (Nikon, Japan) in 10 random fields and described as fold change. The fold changes are calculated as the ratio of the final value in each group to the value in control group at day 30 (set as “1”).

### 4.10. Statistical Analysis 

Statistical analyses were performed using GraphPad Prism software (ver. 8.0; GraphPad Software Inc., La Jolla, CA, USA). The Kruskal–Wallis test followed by post hoc testing with Dunn’s test was used to examine differences between groups. In all analyses, *P* < 0.05 was taken to indicate statistical significance.

## Figures and Tables

**Figure 1 ijms-21-04136-f001:**
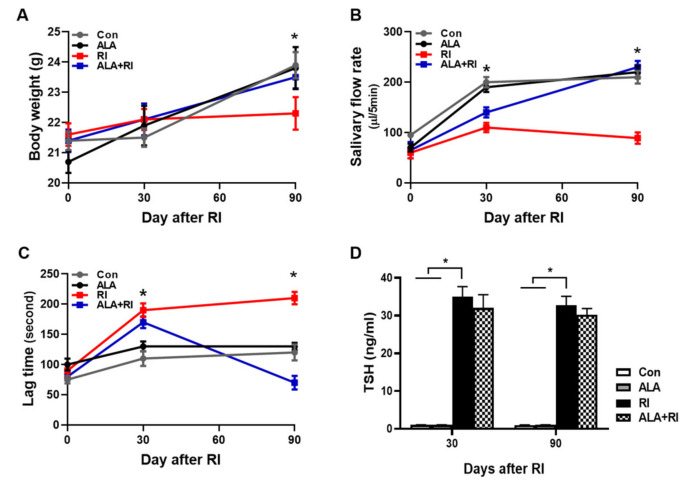
Changes of mouse body weights, salivary flow rate, and lag times after radioiodine (RI) exposure. Mice were subjected to radioiodine (RI, 0.01 mCi/g body weight orally). ALA was administered 24 h 30 min before RI exposure (100 mg/kg body weight, i.p.). (**A**) Body weight, (**B**) salivary flow rate, and (**C**) saliva lag time were measured at each time points after RI exposure. Scoring of salivary flow rate and lag time is described in the “Materials and Methods” section (B, C) The ALA-treated group showed improved lag time and increased saliva secretion relative to the RI only group. (**D**) To verify the effects of ALA on thyroid function following RI therapy, ELISAs for serum TSH were performed (both 30 and 90 days). The serum TSH levels were significantly increased in RI only group and tend to decrease by ALA treatment but not significant. The Kruskal–Wallis test followed by post hoc testing with Dunn’s test was used to examine differences between groups. In all analyses, * *P* < 0.05 was taken to indicate statistical significance.

**Figure 2 ijms-21-04136-f002:**
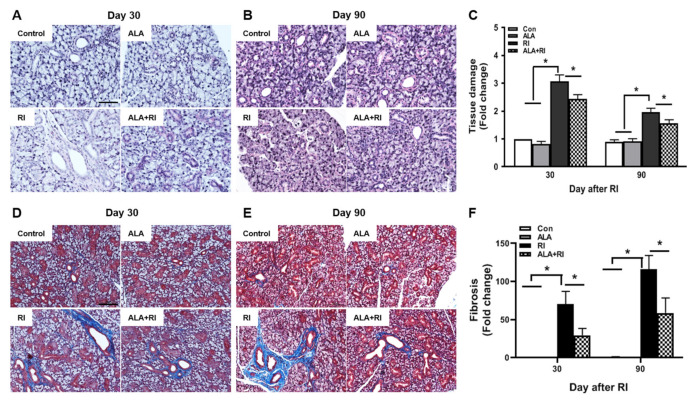
Pathological findings following radioiodine (RI) therapy. (**A**–**C**) Representative morphological images of acinar and ductal cells in the salivary glands (SGs). Control and alpha lipoic acid (ALA) glands showed preserved structure of the glands. RI-treated glands showed severe tissue injury. (**D**–**F**) Representative Masson’s trichrome-stained sections of the SGs. Interstitial and vascular fibrosis in the SGs was minimal in the control and ALA groups, but was significantly greater in the RI only group than in the RI + ALA group. (**A**,**D**) 30 days post-RI therapy; (**B**,**E**) 90 days post-RI therapy. Data are means ± SD. * *P* < 0.05 compared to the indicated groups. Scale bars: 100 μm.

**Figure 3 ijms-21-04136-f003:**
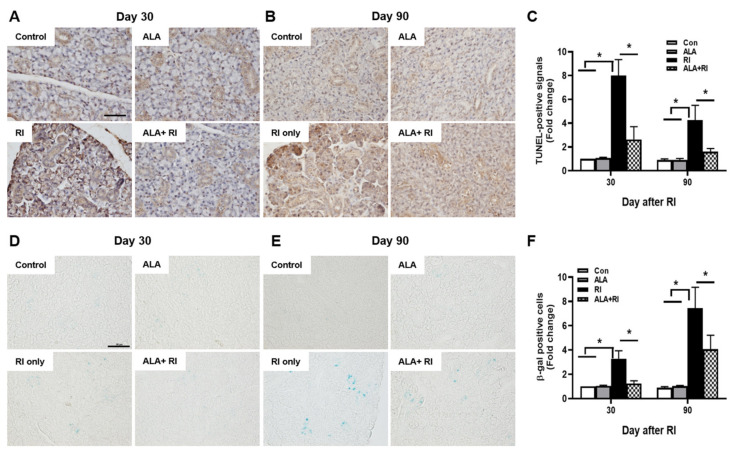
Increased cellular apoptosis and senescence after RI therapy. Tissues were dissected from each mouse, fixed in 4% paraformaldehyde, and cut into sections 5 μm thick. (**A**–**C**) Apoptosis was examined by terminal deoxynucleotidyl transferase biotin-dUTP nick end labeling (TUNEL) assay. Positive signals were detected in the nuclei of the acinar and ductal cells in the RI only group and signal intensity was greater at 30 days than 90 days post-RI therapy. Serial sections were assessed for β-galactosidase activity by staining with X-gal (blue) (**D**–**F**). Note the appearance of β-galactosidase staining in the acinar and ductal cells from the RI only group. (**A**,**D**) 30 days post-RI therapy; (**B**,**E**) 90 days post-RI therapy. Data are means ± SD. * *P* < 0.05 compared to the indicated groups. Scale bars: 100 μm.

**Figure 4 ijms-21-04136-f004:**
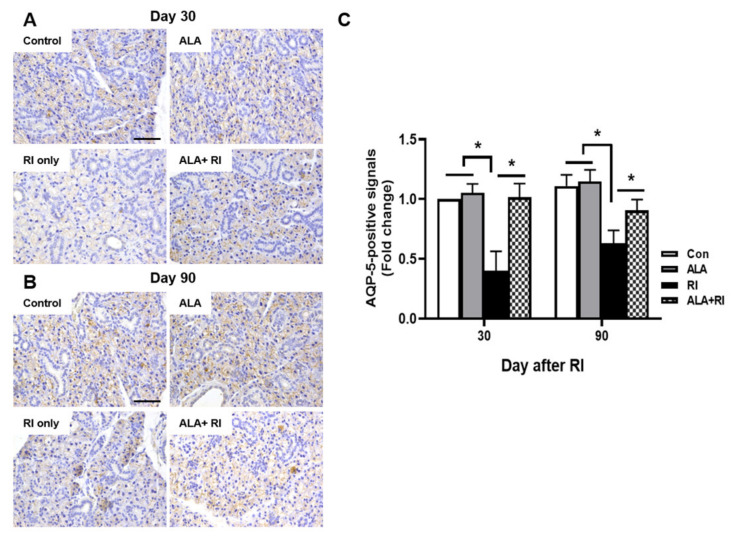
Aquaporin 5 (AQP-5) expression in the SGs of mice that underwent RI therapy. Tissues were dissected from each mouse, fixed in 4% paraformaldehyde, and cut into sections 5 μm thick. Diaminobenzidine (DAB) staining (brown) of AQP-5 to identify salivary function. Representative sections immunohistochemically stained for AQP-5 expression in the acinar cells of the control and ALA only groups. AQP-5 expression level was significantly higher in the RI + ALA group than the RI only group. (**A**,**C**) 30 days post-RI therapy; (**B**,**C**) 90 days post-RI therapy. Data are means ± SD. * *P* < 0.05 compared to the indicated groups. Scale bars: 100 μm.

**Figure 5 ijms-21-04136-f005:**
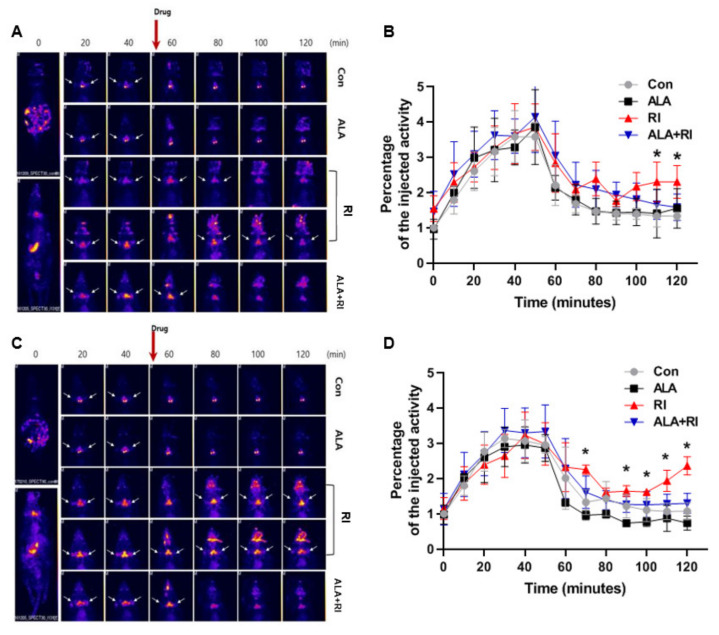
Planar whole-body images and dynamics of 99mTc pertechnetate uptake and excretion. (**A**,**C**) Representative single-photon emission computed tomography (SPECT)-scans of mice that underwent RI therapy. (**B**,**D**) The rates of 99mTc pertechnetate uptake were similar among all mice both 30 and 90 days after therapy. The level of excretion was significantly lower in the RI only group than the RI + ALA group, and the differences in excretion were greater at 90 days than at 30 days. (**A**,**B**) 30 days post-RI therapy; (**C**,**D**) 90 days post-RI therapy. Data are means ± SD. * *P* < 0.05 compared to the RI only and RI + ALA groups. The drug is pilocarpine.

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
