# Peer review of "Protective Effect of Alpha-Lipoic Acid on Salivary Dysfunction in a Mouse Model of Radioiodine Therapy-Induced Sialoadenitis"

_ijms, 2020, doi:10.3390/ijms21114136_

Round 1

Reviewer 1 Report

This manuscript reports a study about the radioprotective effects of alpha lipoic acid on radioiodine therapy in a model mouse. The alpha lipoic acid was administere pior to radioiodine therapy which is known to cause salivary gland (SG) dysfunction The manuscript could be accepted after minor revisions as followings. 1) Some abbreviations should be defined the first time that it appears in the text. Some examples are the following: Page 4, heading 2.2., first line, TSH should be defined Page 4, heading 2.3., H&E y MT should be defined Page 5, heading 2.4., TUNNEL should be defined Page 5, heading 2.7., SPECT should be defined 2) A figure with the chemical structure of alpha lipoic acid should be included

Reviewer 2 Report

In this manuscript, the authors describe an investigation of the effects of alpha-lipoic acid (ALA) on radioiodine therapy-induced salivary gland dysfunction in mice. The study addresses a prevalent clinical issue in RI and the rationale is well-substantiated based on ALA antioxidant properties and current literature. The manuscript is well-written and appropriate context is provided in language comprehensible to broad range of readers.

Mice were either untreated, treated with ALA alone, RI alone, or a combination thereof. The authors present salivary gland assessment at both 30 and 90 days following treatment, including salivary flow, excretion kinetics by SPECT imaging, TSH measurements, and histological approaches. I commend the authors on investigating the effects on SG function using a wide variety of readouts. RI mice pre-treated with ALA had significantly improved salivary flow and lag time, less fibrosis and apoptosis, and better tracer excretion when compared to the RI-only group. The authors conclude that ALA is a potentially viable salivary gland radioprotectant for application in RI therapy.

Specific comments:

  • Clarify throughout the manuscript that the ALA was administered both at 24 h and 30 minutes prior to RI (not a single 24.5 h time point prior)
  • Were male or female mice used? (Abstract states use of female mice; Method states males)
  • Figure 1A-C: add standard deviation and clarify between which groups the statistically significant differences were achieved.
  • Results: “The level of 99mTc pertechnetate was significantly low in the RI only group at 110 and 120 minutes.”. To match with the corresponding Figure 5B, I believe this should be corrected to reflect that the degree of elimination is lesser.
  • Discussion: Delete the following repeated sentence: “Therefore, the present study was performed to investigate the radioprotective effects of ALA against RI-induced SG injury.”
  • Discussion: “We verified that the thyroid glands were completely ablated 7 days after RI administration by SPECT.” Provide information regarding SPECT acquisition and image analysis.
  • Figure 5A,C: Are the two rows of RI referring to two different RI-only mice? Clarify in the Figure 5 caption that the drug indicated in the figure is the pilocarpine-induced excretion.

Questions/Additional experiments:

  • What about the effect of ALA on malignant tissue? Is it sufficient for them to show elevated TSH in the RI groups and no thyroid on SPECT at 7 days to say the ablation in a malignant thyroid would be unaffected by ALA? I think in general the manuscript would benefit from some kind of investigation into the effect of ALA on ablation dynamics/efficacy, but clearly that would be a significant additional experiment i.e. orthotopic thyroid cancer cell injection into thyroid gland and monitoring after ALA + RI..
  • Question about the SPECT figures.. is the active region eg. in first row at 60min the thyroid? And I guess the SG ROIs are just on either side which is used for the excretion curves? Wouldn’t we not see the thyroid in the RI/ALA+RI groups? Or maybe that is not the thyroid..?
  • It would have been interesting to see if the effects of ALA are dose-dependent. How was the 100mg/kg dose selected?
  • The authors state that surgical approaches for mouse model thyroidectomy are difficult because they are often incomplete. Wouldn’t such a model, that is possibly incomplete followed by RI for remnant ablation be somewhat similar to the clinical reality?

Reviewer 3 Report

In their manuscript Dr. Jung and colleagues evaluated the protective effect of the alpha-lipoic acid on salivary gland following radioiodine treatment in male C57NL/6 mice. Although the reported findings are of some interest they lack of novelty since very similar information have been reported by the same authors in a previous publication on a rat model (Oncotarget 7:15105-15117, 2016). Besides that, there is a number of issues, below reported that should be addressed by the Authors.

Comments

  1. In several instances, English presentation could be sensibly improved.
  2. The main aim of radioiodine treatment is to eliminate thyroid cancer cells. The reported ability to attenuate the cytotoxic effects of radioiodine, not only on salivary gland but also on thyroid cells, as reported by the same authors in a previous publication (PLoS One 9:e112253, 2014), should lead to a reduced efficacy of radioiodine therapy in cancer patients. Is it thus worthwhile to use the alpha-lipoic acid to decrease radioiodine therapy? The Authors should appropriately discuss this point.
  3. Authors should show the expression of oxidative stress markers in their experimental settings, and not to refer to previous publications on different experimental animal models.
  4. In addition, the experimental doses of the alpha-lipoic acid should be confirmed in their mouse experimental system and not simply refer to previous publications performed on different experimental settings.
  5. It is also not clear on what bases the timing to evaluate the protective effects of the alpha-lipoic acid (30 and 90 days) were chosen. In previous publications by the same authors the analyze the effects of the alpha-lipoic acid after 4 and 7 days or 4, 7, 28 and 56 days. Please, provide a rationale for the timing used in the present study.
  6. In the Materials and Methods section, please, specify that the kit used for TSH assay was a mouse TSH assay. In addition, I would suggest including the analysis of serum thyroid hormone levels, beside that of TSH.
  7. In figure 3, 4 and 5 a better identification of which pictures are related to the 30 days, and which at the 90 days will be helpful.

Round 2

Reviewer 3 Report

The authors have addressed all my concerns and I do not have any further comments.